# Predicting Mortality for COVID-19 Patients Admitted to an Emergency Department Using Early Warning Scores in Poland

**DOI:** 10.3390/healthcare12060687

**Published:** 2024-03-19

**Authors:** Patryk Rzońca, Sławomir Butkiewicz, Paula Dobosz, Artur Zaczyński, Marcin Podgórski, Robert Gałązkowski, Waldemar Wierzba, Katarzyna Życińska

**Affiliations:** 1Department of Human Anatomy, Faculty of Health Sciences, Medical University of Warsaw, 02-091 Warsaw, Poland; 2Emergency Department, The National Institute of Medicine of the Ministry of Interior and Administration, 02-507 Warsaw, Poland; slawomir.butkiewicz@cskmswia.gov.pl; 3Institute of Genetics and Biotechnology, Faculty of Biology, University of Warsaw, 00-927 Warsaw, Poland; 4Clinical Department of Neurosurgery, The National Institute of Medicine of the Ministry of Interior and Administration, 02-507 Warsaw, Poland; artur.zaczynski@cskmswia.gov.pl; 5Department of Emergency Medical Services, Faculty of Health Sciences, Medical University of Warsaw, 02-091 Warsaw, Poland; marcin.podgorski@wum.edu.pl (M.P.); robert.galazkowski@wum.edu.pl (R.G.); 6Satellite Campus in Warsaw, University of Humanities and Economics, 90-212 Łódź, Poland; waldemar.wierzba@cskmswia.gov.pl; 7Department of Rheumatology, Systemic Connective Tissue Diseases and Rare Diseases, The National Institute of Medicine of the Ministry of Interior and Administration, 02-507 Warsaw, Poland

**Keywords:** COVID-19, mortality, early warning score, emergency department, outcome

## Abstract

COVID-19 disease is characterised by a wide range of symptoms that in most cases resemble flu or cold. Early detection of infections, monitoring of patients’ conditions, and identification of patients with worsening symptoms became crucial during the peak of pandemic. The aim of this study was to assess and compare the performance of common early warning scores at the time of admission to an emergency department in predicting in-hospital mortality in patients with COVID-19. The study was based on a retrospective analysis of patients with SARS-CoV-2 infection admitted to an emergency department between March 2020 and April 2022. The prognostic value of early warning scores in predicting in-hospital mortality was assessed using the receiver operating characteristic (ROC) curve. Patients’ median age was 59 years, and 52.33% were male. Among all the EWS we assessed, REMS had the highest overall accuracy (AUC 0.84 (0.83–0.85)) and the highest NPV (97.4%). REMS was the most accurate scoring system, characterised by the highest discriminative power and negative predictive value compared to the other analysed scoring systems. Incorporating these tools into clinical practice in a hospital emergency department could provide more effective assessment of mortality and, consequently, avoid delayed medical assistance.

## 1. Introduction

The SARS-CoV-2 virus was first identified in late December 2019 in Wuhan, China. It is a pathogen belonging to the coronavirus family that spreads through droplet or contact transmission [1,2]. In humans, it causes a disease called COVID-19, which is characterised by a wide range of symptoms that in most cases resemble flu or cold. However, in some cases, it can lead to serious complications, including pneumonia, respiratory failure, and death [2,3]. The severity of the disease and mortality due to COVID-19 may vary depending on the patient’s age, health status, presence of comorbidities, and regional availability of medical services. According to numerous studies, about 15% of patients infected with SARS-CoV-2 require hospitalisation, with approximately 5% requiring intensive care unit treatment [4,5,6,7,8].

The COVID-19 pandemic had a huge impact on healthcare systems worldwide, forcing quick and effective action to combat the disease. In the initial phase of the pandemic, the biggest challenge was the rapid increase in the number of patients requiring treatment, including hospitalisation, mechanical ventilation, and intensive care. The pressure that healthcare systems faced when COVID-19 became a global problem was enormous and necessitated many changes aimed at dealing with a large number of patients infected with SARS-CoV-2 [9,10,11]. The methods for dealing with this problem varied depending on the country and region. Many countries decided to build new hospitals, usually temporarily kept, to accommodate a large number of patients, while others, operating under crisis conditions, decided to transform existing infrastructure into temporary hospitals. Ensuring adequate infrastructure, medical equipment, and appropriate healthcare staff was just the first step taken in the event of an increasing number of COVID-19 patients [12,13,14].

During the COVID-19 pandemic, early detection of infections, monitoring of patients’ conditions, and identification of patients with worsening symptoms became crucial. To achieve this goal, many countries utilised existing early warning scores (EWS) that are widely used in clinical practice in hospital emergency departments [15]. The use of EWS became particularly significant during the pandemic, as they enabled the rapid identification of patients requiring medical intervention as well as assessments of the risk of hospitalisation and the need for treatment in intensive care units. Moreover, EWS scales allow for the monitoring of COVID-19 patients and determining whether their condition is improving or worsening and provide a tool to support physicians in making therapeutic decisions and planning the treatment of COVID-19 patients [16,17,18,19]. The use of EWS scales may also play a key role in making decisions about which patients should be admitted to the hospital emergency department and receive treatment in the hospital setting and which can be discharged for outpatient treatment. This aspect of patient management is particularly important in light of a recent study by Fried et al., who demonstrated that with correct patient selection, even severe COVID-19 cases can be safely treated in the outpatient setting [20].

The aim of this study was to assess and compare the performance of the common early warning scores at the time of admission to the emergency department in predicting in-hospital mortality in patients with COVID-19.

## 2. Materials and Methods

### 2.1. Study Design and Setting

This was a retrospective study carried out in a National Institute of Medicine of the Ministry of Interior and Administration in Warsaw (Poland), which is one of the largest teaching hospitals in Warsaw and was a Referral Centre for COVID-19 in central Poland. The study was carried out based on the medical records covering the period between March 2020 and April 2022 of patients admitted to the Emergency Department of the Central Clinical Hospital of the Ministry of Interior and Administration in Warsaw.

### 2.2. Study Cohort and the Eligibility Criteria

A total cohort of 5024 cases were included in the final analysis. In the following study, all adult (>18 years old) patients admitted to the Emergency Department with SARS-CoV-2, which was confirmed by laboratory testing (dedicated test performed with the real-time reverse-transcriptase-polymerase-chain-reaction technique—RT-PCR) and/or whose final diagnosis expressed with the aid of the ICD-10 (International Classification of Diseases 10) code was U07.1, meaning COVID-19 disease. The exclusion criteria were: pregnancy among women (*n* = 316), pneumonia caused by other pathogens (*n* = 713), and incomplete data required to retrospectively calculate early warning scores (*n* = 1301).

### 2.3. Data Collection and Measurements

The data were extracted from the hospital’s internal database of clinical records and prepared by the hospital’s IT Department. Extracted data included: age and sex of the patients, length of hospital stay, temperature, heart rate (HR), respiratory rate (RR), systolic blood pressure (SBP), diastolic blood pressure (DBP), mean arterial pressure (MAP), laboratory test results, Glasgow Coma Scale (GCS) score, oxygen therapy, and peripheral oxygen saturation (SpO_2_), clinical symptoms and comorbidities, as well as outcomes of hospitalisation (survival or death).

Among all the EWSs available, we chose those that were the quickest to use and that could be calculated for each patient from available vital signs and physiological measurements recorded on admission to the Emergency Department. In this study, we selected six EWSs: Modified Early Warning Score (MEWS) [21], National Early Warning Score 2 (NEWS2) [22], National Early Warning Score (NEWS) [23,24], Standardised Early Warning Score (SEWS) [25], Rapid Emergency Medicine Score (REMS) [26], and Rapid Acute Physiology Score (RAPS) [27]. During the pandemic, MEWS was used to identify patients admitted to the hospital emergency department, and its results were featured in the database we obtained. The scores on the other scales, on the other hand, were calculated retrospectively. For NEWS2 calculation, patients were considered at risk of hypercapnic respiratory failure (SpO_2_ Scale 2) if they had a confirmed history of chronic obstructive pulmonary disease (COPD).

### 2.4. Statistical Analysis

Categorical variables were reported as numbers (*n*) and percentages (%), while continuous variables were reported as medians (Me) and interquartile ranges (IQR). Data distribution was evaluated using the Kolmogorov–Smirnov test and the Lilliefors test. Baseline data were compared using the chi-squared test for categorical variables and the Mann–Whitney U-test for continuous variables. 

The prognostic value of early warning scores in predicting in-hospital mortality was assessed using the receiver operating characteristic (ROC) curve. The optimal cut-off values were calculated by the Youden index. Sensitivity, specificity, positive predictive value (PPV), negative predictive value (NPV), positive likelihood ratio (+LR), and negative likelihood ratio (−LR) were then calculated. The areas under the ROC curve (AUROCs) were compared by the method described by DeLong.

The data obtained were analysed statistically using IBM SPSS statistics for Windows, Version 25.0 (Armonk, NY, USA: IBM Corp.) and MedCalc Statistical Software version 20.218 (MedCalc Software Ltd., Ostend, Belgium). In the study, a two-tailed *p* value of <0.05 was considered statistically significant.

### 2.5. Ethics

The study protocol was approved by the Bioethics Committee at the Medical University of Warsaw, which confirmed that the study did not require consent due to its retrospective nature (AKBE/13/2022). The study was performed in accordance with the principles established in the 1964 Declaration of Helsinki and its later amendments. Reports from the database did not permit identification of individual patients at any stage of the study.

## 3. Results

### 3.1. Population and Outcomes

Baseline characteristics of all 5024 patients who were included in the study cohort are presented in Table 1. Patients’ median age was 59 (IQR 42–74) years and 52.33% (2629 patients) were male. More than 21% of the patients required passive oxygen therapy, 1.13% patients required mechanical ventilation and more than 18% of the patients required nasal high-flow therapy. Admission to the ICU was recorded for more than 6% of the patients. The most commonly reported comorbidities were hypertension (23.09%) and diabetes (10.27%). Fever was the most common symptom (26.21%), followed by dyspnoea (18.31%) and cough (17.54%). The baseline MEWS, SEWS, NEWS, NEWS2, REMS and RAPS at the moment of admission were 1 (0–3), 1 (0–2), 2 (0–4), 2 (0–4), 5 (2–7), and 0 (0–2), respectively. Detailed results are shown in Table 1.

Cases were categorised into non-survivor and survivor groups. Patients in the non-survivor group were more likely to be male (61.64% vs. 50.98%), of older age (median 77 vs. 55 years). The comorbidities identified were more common among patients who did not survive. Fever (31.29% vs. 25.29%) and dyspnoea (28.93% vs. 16.77%) were more common in the non-survivor group. Passive oxygen therapy (54.09% vs. 17.05%), ventilator therapy (7.55% vs. 0.21%), nasal high-flow therapy (41.35% vs. 14.86%) and intensive care unit admission (32.08% vs. 2.42%) were more often required in the non-survivor group. The survivor and non-survivor groups also differed in terms of the vital signs and laboratory test results. Non-survivors were more likely to have higher MEWS (3 vs. 1), SEWS (3 vs. 1), NEWS (5 vs. 1), NEWS2 (5 vs. 2), REMS (8 vs. 4) and RAPS (1 vs. 0) (Table 1).

### 3.2. Prognostic Accuracy of Early Warning Score in Predicting the in-Hospital Mortality Rate

To assess the utility of EWS to predict the in-hospital mortality, the ROC curves were constructed and the AUCs were calculated (Figure 1A). The AUCs of MEWS, NEWS2, NEWS, SEWS, REMS, and RAPS were 0.74, 0.74, 0.79, 0.77, 0.84, and 0.66, respectively. Based on the best Youden index, an optimum cut-off value was used to predict in-hospital mortality using each score. The cut-off values for each score, together with the sensitivity, specificity, positive predictive value (PPV), negative predictive value (NPV), positive likelihood ratio (+LR), and negative likelihood ratio (−LR) are shown in Table 2. Among all the EWS we assessed, REMS had the highest overall accuracy (AUC 0.84 (0.83–0.85)) and the highest NPV (97.4%) (Table 2).

In the subgroup of age <65 years, the AUCs of MEWS, NEWS2, NEWS, SEWS, REMS, RAPS were 0.76, 0.77, 0.81, 0.79, 0.81, and 0.71, respectively (Figure 1B), whereas in the subgroup of age ≥65 years, the AUCs of MEWS, NEWS2, NEWS, SEWS, REMS, RAPS were 0.69, 0.68, 0.72, 0.72, 0.69 and 0.62, respectively (Figure 1C). Table 2 shows the cut-off values for each score based on the best Youden index together with the sensitivity, specificity, positive predictive value (PPV), negative predictive value (NPV), positive likelihood ratio (+LR), and negative likelihood ratio (−LR). An optimum cut-off value was used to predict in-hospital mortality using each score appropriate for the subgroup of age ≥65 years and the subgroup of age <65 years. Among all the EWS we assessed in the subgroup of age <65 years, NEMS had the highest overall accuracy (AUC 0.81 (0.80–0.82)) and the highest sensitivity (81.6%) and NPV (99.2%). In the subgroup of age ≥65 years, SEWS had the highest overall accuracy (AUC 0.72 (0.70–0.74)) and the highest sensitivity (71.4%) and NPV (85.8%) (Table 2).

Pairwise comparisons of the AUCs associated with the six EWSs showed significant differences among all investigated scales. The AUC of the REMS for predicting in-hospital mortality was much higher than that for MEWS, NEWS2, NEWS, SEWS, REMS, RAPS. The greatest difference between areas under curves, respectively, was observed in the case of REMS—RAPS (0.188, <0.001), REMS—MEWS (0.103, <0.001) and REMS—NEWS2 (0.102, <0.001) (Table 3).

Pairwise comparisons of the AUCs associated with the 6 EWSs showed significant differences among 9 pairs in the subgroup of age <65 years and 11 pairs in the subgroup of age ≥65 years. In the subgroup of age <65 years, the AUC of the NEMS for predicting in-hospital mortality was much higher than that for MEWS, NEWS2, SEWS, REMS, RAPS. The greatest difference between areas under curves was observed between NEMS and SEWS (0.105, <0.001). In the subgroup of age ≥65 years, the AUC of the SEWS for predicting in-hospital mortality was much higher than that for MEWS, NEWS2, NEWS, REMS, RAPS, of which the greatest difference between areas under curves was observed between SEWS and RAPS (0.106, <0.001) (Table 3).

## 4. Discussion

The SARS-CoV-2 pandemic, its impact, and consequences have been the subject of numerous scientific publications worldwide. The aims of these studies have been and continue to be the analysis of risk factors for severe disease progression and in-hospital mortality, diagnostic evaluation, and treatment; thus, ongoing research is necessary for a better understanding of this global problem. This requires patience, improvement, innovation, the need for evolution, and learning from others’ experience during the SARS-CoV-2 pandemic [28,29,30,31,32]. Therefore, the aim of the undertaken research was to assess and compare the performance of common early warning scores (EWS) at the time of admission to the Accident and Emergency department in predicting in-hospital mortality in patients with COVID-19.

The results of our research have shown that, among the analysed participants who did not survive, there were males and older individuals, which is consistent with the findings of López-Pérez et al. [33]. At the same time, the literature emphasises strongly that advanced age and male gender are the main risk factors for severe disease and mortality due to COVID-19 [32,33,34]. Numerous studies published to date have demonstrated that comorbidities such as hypertension, diabetes, COPD, malignancy and chronic kidney disease are associated with a severe course of infection and higher mortality due to COVID-19 [35,36,37,38,39,40,41]. Our results have shown that the highest mortality rate was associated with hypertension, renal failure and diabetes. Furthermore, the main symptoms of COVID-19 include fever, headache, muscle pain, cough, fatigue, and shortness of breath [28,34,42,43]. In our study, non-survivors more frequently required specialised medical procedures such as respiratory therapy and nasal high-flow therapy. They also more frequently required admission to the Intensive Care Unit. Similar results were obtained by van Halem et al. and Díaz-Vélez et al. [44,45]. The clinical picture in hospitalised patients who died due to COVID-19 showed elevated values of selected vital signs, such as heart rate and respiration rate, and lower MAP or oxygen saturation values. Moreover, the results of laboratory tests pointed to higher inflammation, which has been corroborated by our findings and those of other researchers [32,45,46,47]. The obtained results of our research reflect scientific reports from around the world regarding the characteristics of individuals who did not survive due to COVID-19.

It should be emphasised that since the beginning of the COVID-19 pandemic, a total of 6,950,655 deaths have been reported worldwide, according to data from the World Health Organization [48]. Rentsch et al. demonstrated a mortality rate of 10.7% among their studied patients with a positive PCR test in the United States [49]. On the other hand, Olivas-Martinez et al. reported an in-hospital mortality rate of 30.1% for hospitalised patients with severe COVID-19 in a tertiary care centre in Mexico [50]. Aygun et al. found a 16.8% 28-day mortality rate among their analysed patients with a positive PCR test in Turkey [17]. López-Pérez et al. reported a mortality rate of 13.1% among the analysed patients in their study on risk factors for mortality of hospitalised adult patients with COVID-19 pneumonia in a private tertiary care centre in Mexico [33]. In our analysis, the mortality rate among patients was nearly 22%. It should be noted that factors such as the duration of the study period and the organisation of the healthcare system (including the level of the referring hospital) can influence the mortality rates observed in COVID-19 patients.

A study by Martín-Rodríguez et al. on patients transferred to the hospital emergency department with suspected COVID-19 infection showed that non-survivors had higher scores on the following scales: NEWS2, qSOFA, Modified REMS and RAPS [51]. In our study, patients who did not survive also had significantly higher scores on early warning scales. Therefore, the identification of patients who are likely to deteriorate clinically and die from COVID-19 in a short period of time is absolutely essential for proper organisation of work in the hospital emergency department and rational utilisation of medical staff. Of all the early warning scores analysed, the best overall prognostic performance was obtained by the Rapid Emergency Medicine Score (REMS), which had the highest discriminatory power (AUC 0.84) and the highest negative predictive value (97.4%) when assessed upon admission to the emergency department compared to other scoring systems. This is consistent with studies by other researchers [27,52,53]. These results can be explained by the components of REMS, i.e., MAP and age, which are not part of the other EWS scales analysed. A study by Nam et al. on the effect of blood pressure variability in patients with COVID-19 and hypertension showed that age and higher MAPcv were significantly associated with in-hospital mortality [54]. It is worth emphasising that there are also studies that used other early warning scales. Wei et al. found that the National Early Warning Score 2 (NEWS2) exhibits excellent sensitivity and specificity in predicting early mortality in both prehospital and emergency department settings. In their study comparing triage tools for identifying mortality risk and injury severity in patients with multiple traumas admitted to the emergency department (ED) during daytime and night time [55], Ying et al. demonstrated that NEWS is the best tool in this regard for both daytime and night-time admissions. They also found that the modified Rapid Emergency Medicine Score (mREMS) was better at identifying serious injuries during the day [56]. Kostakis et al. conducted an analysis of the use of NEWS and NEWS2 in hospitalised patients with SARS-CoV-2 and found that the results of NEWS or NEWS2 were good and similar across all five analysed cohorts (range = 0.842–0.894), suggesting that adjustments to these scores, such as adding new variables or changing the weights of existing parameters, are not necessary for evaluating patients with COVID-19 [57].

Several studies indicate that advanced age is an independent factor associated with mortality in patients with COVID-19 [58,59,60,61]. Therefore, the next step of our study was to calculate the prognostic value of early warning scales in two subgroups (<65 years and ≥65 years). Analysis of our research revealed that for the subgroup of patients under the age of 65, NEWS and REMS upon admission to the emergency department had the highest discriminatory power values (AUC 0.81) compared to other scoring systems. For the subgroup of patients aged 65 and above, NEWS and Standardised Early Warning Score (SEWS) achieved the highest discriminatory power values (AUC 0.72) compared to the other scores. Additionally, the findings of Hu et al. showed that NEWS and NEWS2 had the highest AUC values (0.829) for the subgroup aged 65 and above, while for the subgroup under the age of 65, SEWS had the highest AUC (0.893) [15].

This study had several limitations. Firstly, it was a single-centre study and had a retrospective nature. Further validation in a multi-centre cohort is still required. Secondly, in the study, the EWS value was calculated only based on parameters at the time of admission to the hospital emergency department, without recording parameter changes during the hospital stay. Thirdly, the primary outcome measure was in-hospital mortality, assuming that all patients who were discharged did not die. Despite these limitations, we made every effort to ensure the study had high-quality results.

## 5. Conclusions

REMS was the most accurate scoring system, characterised by the highest discriminative power and negative predictive value compared to the other analysed scoring systems. This may stem from the fact that REMS is the only scale whose parameters include the patient’s age and MAP. In the group of patients below 65 years of age, NEWS and REMS were the most effective in predicting mortality in COVID-19 patients, while in the group of patients aged 65 years and above, NEWS and SEWS had the highest predictive value. Despite MEWS being used in many hospitals during the pandemic in Poland, it is not suitable for COVID-19 patients, as its performance and prognostic value in predicting mortality are inferior to other early warning scales. Incorporating these tools into clinical practice in the hospital emergency department could provide more effective assessment of mortality and, consequently, avoid delayed medical assistance.

## Figures and Tables

**Figure 1 healthcare-12-00687-f001:**
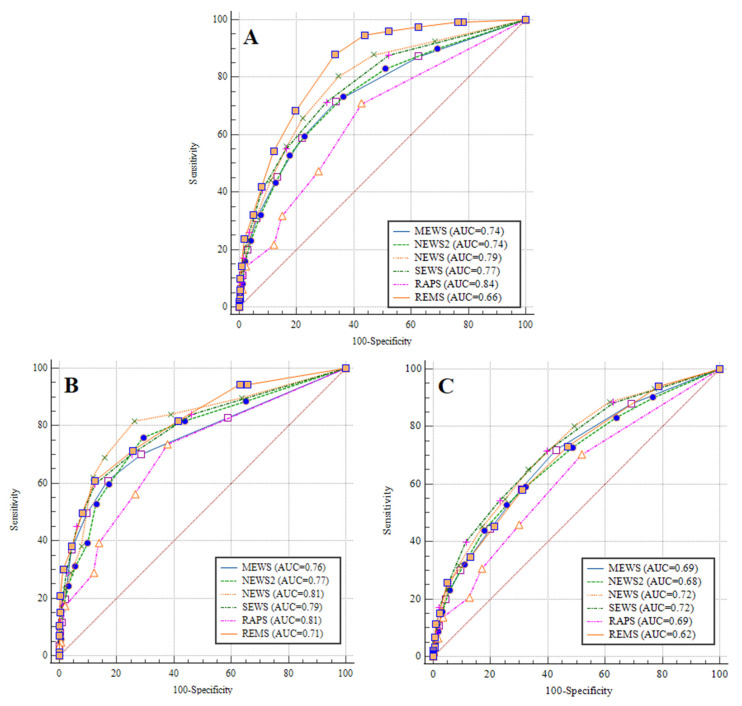
Receiver operator characteristic curves illustrating the ability of MEWS score, NEWS2 scores, NEWS score, SEWS score, REMS score, RAPS score at emergency department admission to predict mortality from COVID-19 for overall cases and for subgroups ((**A**)—overall cases; (**B**)—subgroup of age <65 years; (**C**)—subgroup of age ≥65 years).

**Table 1 healthcare-12-00687-t001:** Patients’ characteristics and a comparative analysis between the surviving and non-surviving groups of patients with COVID-19 infection.

Variables	Total (*n* = 5024)	Survivors (*n* = 4023)	Non-Survivors (*n* = 1104)	*p*-Value
Age [years]—Me (IQR)	59 (42–74)	55 (40–70)	77 (69–86)	<0.001
>65—*n* (%)	2122 (42.24)	1573 (35.85)	549 (86.32)	<0.001
Sex [Male]—*n* (%)	2629 (52.33)	2237 (50.98)	392 (61.64)	<0.001
Comorbidities—*n* (%)				
Hypertension	1160 (23.09)	877 (19.99)	283 (44.50)	<0.001
Diabetes	516 (10.27)	360 (8.20)	156 (24.53)	<0.001
Acute coronary syndromes history	367 (7.30)	256 (5.83)	111 (17.45)	<0.001
Stroke history	254 (5.06)	175 (3.99)	79 (12.42)	<0.001
Cancer	202 (4.02)	127 (2.89)	75 (11.79)	<0.001
Renal failure	422 (8.40)	261 (5.95)	161 (25.31)	<0.001
Heart failure	350 (6.97)	237 (5.40)	113 (17.77)	<0.001
COPD	107 (2.16)	72 (1.64)	35 (5.50)	<0.001
Smoking	248 (4.94)	197 (4.49)	51 (8.02)	<0.001
Symptoms—*n* (%)				
Fever	1317 (26.21)	1118 (25.48)	199 (31.29)	0.002
Cough	881 (17.54)	768 (17.50)	113 (17.77)	0.870
Dyspnoea	920 (18.31)	736 (16.77)	184 (28.93)	<0.001
Muscle pain	365 (7.27)	339 (7.73)	26 (4.09)	0.001
Diarrhoea	361 (7.19)	309 (7.04)	52 (8.18)	0.301
Loss/change in sense of taste	271 (5.39)	254 (5.79)	17 (2.67)	0.001
Loss/change in sense of smell	311 (6.19)	291 (6.63)	20 (3.14)	0.001
Headache	386 (7.68)	362 (8.25)	24 (3.77)	<0.001
Passive oxygen therapy [Yes]—*n* (%)	1092 (21.74)	748 (17.05)	344 (54.09)	<0.001
Ventilator therapy [Yes]—*n* (%)	57 (1.13)	9 (0.21)	48 (7.55)	<0.001
Nasal high-flow therapy [Yes]—*n* (%)	915 (18.21)	652 (14.86)	263 (41.35)	<0.001
ICU admission [Yes]—*n* (%)	310 (6.17)	106 (2.42)	204 (32.08)	<0.001
Vital signs—Me (IQR)				
Systolic blood pressure [mmHg]	133 (119–149)	134 (120–149)	127 (108–150)	<0.001
Diastolic blood pressure [mmHg]	81 (72–90)	81 (93–90)	75 (63–88)	<0.001
MAP [mmHg]	99 (89–108)	99 (90–108)	93 (81–107)	<0.001
Heart rate [beats per minute]	87 (77–100)	87 (76–100)	92 (78–110)	<0.001
Respiratory rate [breaths per minute]	17 (15–20)	17 (15–19)	20 (18–25)	<0.001
Oxygen saturation [%]	97 (95–99)	97 (95–99)	92 (86–96)	<0.001
Body temperature [°C]	36.7 (36.4–37.5)	36.7 (36.4–37.4)	36.9 (36.3–38.0)	0.083
Laboratory test results—Me (IQR)				
WBC count [thousand/µL]	6 (5–9)	6 (5–8)	10 (6–18)	<0.001
RBC count [million/µL]	5 (4–5)	5 (4–5)	4 (3–4)	<0.001
Haemoglobin [g/dL]	14 (12–15)	14 (13–15)	12 (10–13)	<0.001
Hematocrit [%]	40 (37–44)	40 (38–44)	36 (32–40)	<0.001
Platelet count [thousand/µL]	209 (163–265)	210 (165–265)	188 (128–252)	0.001
Neutrophil count [thousand/µL]	4 (3–7)	4 (3–6)	8 (5–14)	<0.001
Lymphocyte count [thousand/µL]	1.3 (0.9–1.8)	1.3 (0.9–1.8)	0.8 (0.5–1.3)	<0.001
Scores on admission—Me (IQR)				
MEWS	1 (0–3)	1 (0–2)	3 (1–5)	<0.001
SEWS	1 (0–2)	1 (0–2)	3 (1–5)	<0.001
NEWS	2 (0–4)	1 (0–3)	5 (3–7)	<0.001
NEWS2	2 (0–4)	2 (0–3)	5 (2–7)	<0.001
REMS	5 (2–7)	4 (2–6)	8 (6–10)	<0.001
RAPS	0 (0–2)	0 (0–2)	1 (0–3)	<0.001

Me—median; IQR—interquartile range; COPD—chronic obstructive pulmonary disease; ICU—intensive care unit; MAP—mean arterial pressure; WBC—white blood cells; RBC—red blood cells; MEWS—Modified Early Warning Score; NEWS—National Early Warning Score; NEWS2—National Early Warning Score 2; SEWS—Standardised Early Warning Score; REMS—Rapid Emergency Medicine Score; RAPS—Rapid Acute Physiology Score.

**Table 2 healthcare-12-00687-t002:** Performance of MEWS score, NEWS2 scores, NEWS score, SEWS score, REMS score, RAPS score in predicting in-hospital mortality of COVID-19 patients.

Score	AUC (95% CI)	*p*-Value	Cut-Off	SEN (%)	SPE (%)	PPV	NPV	LR+	LR−
Overall									
MEWS	0.74 (0.73–0.75)	<0.001	3	71.5	66.3	23.5	94.4	2.12	0.43
NEWS2	0.74 (0.73–0.75)	<0.001	5	73.1	63.7	22.6	94.2	2.0	0.4
NEWS	0.79 (0.77–0.80)	<0.001	5	80.4	65.4	25.2	95.8	2.3	0.3
SEWS	0.77 (0.76–0.78)	<0.001	4	71.2	69.5	25.3	94.3	2.3	0.4
REMS	0.84 (0.83–0.85)	<0.001	7	87.9	66.7	27.7	97.4	2.6	0.2
RAPS	0.66 (0.64–0.67)	<0.001	2	70.8	57.2	19.3	93.1	1.7	0.5
<65 years									
MEWS	0.76 (0.74–0.77)	<0.001	4	60.9	83.1	10.0	98.6	3.6	0.5
NEWS2	0.77 (0.76–0.79)	<0.001	4	75.9	70.7	7.4	99.0	2.6	0.3
NEWS	0.81 (0.80–0.82)	<0.001	4	81.6	73.6	8.7	99.2	3.1	0.3
SEWS	0.79 (0.78–0.81)	<0.001	4	59.8	87.5	12.9	98.6	4.8	0.5
REMS	0.81 (0.80–0.82)	<0.001	6	60.9	87.6	13.2	98.6	4.9	0.5
RAPS	0.71 (0.69–0.72)	<0.001	2	73.6	62.2	5.7	98.7	2.0	0.4
≥65 years									
MEWS	0.69 (0.67–0.71)	<0.001	3	71.8	57.1	36.9	85.3	1.7	0.5
NEWS2	0.68 (0.66–0.70)	<0.001	6	52.6	74.4	41.8	81.8	2.1	0.6
NEWS	0.72 (0.70–0.74)	<0.001	5	65.0	66.8	40.6	84.5	2.0	0.5
SEWS	0.72 (0.70–0.74)	<0.001	3	71.4	60.3	38.5	85.8	1.8	0.5
REMS	0.69 (0.67–0.71)	<0.001	7	57.9	69.0	39.5	82.4	1.9	0.6
RAPS	0.62 (0.60–0.64)	<0.001	2	70.3	48.3	32.2	82.3	1.4	0.6

AUC—area under the curve of the receiver operating characteristic; 95% CI—95% confidence interval; SEN—sensitivity; SPE—specificity; PPV—positive predictive value; NPV—negative predictive value; LR+—likelihood ratio positive; LR−—likelihood ratio negative; MEWS—Modified Early Warning Score; NEWS—National Early Warning Score; NEWS2—National Early Warning Score 2; SEWS—Standardised Early Warning Score; REMS—Rapid Emergency Medicine Score; RAPS—Rapid Acute Physiology Score.

**Table 3 healthcare-12-00687-t003:** The comparisons between AUROCs of EWS for predicting mortality.

**AUROC** **Overall**	**RAPS** **(0.66)**	**REMS** **(0.84)**	**SEWS** **(0.77)**	**NEWS** **(0.79)**	**NEWS2** **(0.74)**
MEWS (0.74)	0.085 ***	0.103 ***	0.031 ***	0.046 ***	0.001 *
NEWS2 (0.74)	0.086 ***	0.102 ***	0.030 ***	0.045 ***	
NEWS (0.79)	0.131 ***	0.057 ***	0.014 **		
SEWS (0.77)	0.117 ***	0.072 ***			
REMS (0.84)	0.188 ***				
**AUROC** **<65 years**	**RAPS** **(0.71)**	**REMS** **(0.81)**	**SEWS** **(0.79)**	**NEWS** **(0.81)**	**NEWS2** **(0.77)**
MEWS (0.76)	0.051 *	0.054 **	0.034 **	0.054 **	0.015 *
NEWS2 (0.77)	0.066 **	0.039 *	0.020 *	0.039 **	
NEWS (0.81)	0.085 **	0.000 *	0.105 ***		
SEWS (0.79)	0.020 *	0.085 **			
REMS (0.81)	0.105 ***				
**AUROC** **≥65 years**	**RAPS** **(0.62)**	**REMS** **(0.69)**	**SEWS** **(0.72)**	**NEWS** **(0.72)**	**NEWS2** **(0.68)**
MEWS (0.69)	0.075 ***	0.001 *	0.032 ***	0.029 ***	0.009 *
NEWS2 (0.68)	0.066 ***	0.010 *	0.041 ***	0.038 ***	
NEWS (0.72)	0.103 ***	0.028 **	0.003 *		
SEWS (0.72)	0.106 ***	0.031 **			
REMS (0.69)	0.075 ***				

*—*p* < 0.05; **—*p* < 0.01; ***—*p* < 0.001. MEWS—Modified Early Warning Score; NEWS—National Early Warning Score; NEWS2—National Early Warning Score 2; SEWS—Standardised Early Warning Score; REMS—Rapid Emergency Medicine Score; RAPS—Rapid Acute Physiology Score.

## Data Availability

The data presented in this study are available on request from the corresponding author.

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
