# Peer review of "Predicting Mortality for COVID-19 Patients Admitted to an Emergency Department Using Early Warning Scores in Poland"

_healthcare, 2024, doi:10.3390/healthcare12060687_

Round 1

Reviewer 1 Report

Comments and Suggestions for Authors

It is a well-conducted study that requires some improvements and clarifications:

Define MEWS, NEWS2, NEWS, SEWS, REMS, RAPS on their first use.

Page 2 line 86 – the verb is missing.

Paragraph 181, page 6 - why did you choose the age of 65? Why didn't you go to table two also on survivors and non-survivors?

Were all the scores performed by you based on the extracted data or at the time of hospitalization?

Discussions

Give up adjectives of strengthening - own

In the comparisons with other studies exclude the year of the study, only the author et al.

References are placed at the end of the sentence.

The discussions should also be completed with the other parameters presented in the results. The focus was only on age and scores (which is normal considering the purpose of the study), but the rest of the identified aspects should be discussed at least briefly.

Several studies indicate that advanced age is an independent factor associated with mortality in patients with COVID-19 - correct, but ethically when you refer to studies that associate age as a risk factor, your own study should not be taken into account.

Some phrases should be rephrased, the duplication rate is quite high.

Author Response

Dear Reviewer,

We wish to thank you very much for the time you have taken to read and review our paper. We hope the changes we have made improved the overall quality of the paper in line with your expectations.

It is a well-conducted study that requires some improvements and clarifications:

Define MEWS, NEWS2, NEWS, SEWS, REMS, RAPS on their first use.

Thank you for this remark. We have defined the abbreviations following the Reviewer’s suggestion.

Page 2 line 86 – the verb is missing.

Thank you for this comment. We have corrected the sentence.

Paragraph 181, page 6 - why did you choose the age of 65? Why didn't you go to table two also on survivors and non-survivors?

Thank you for this comment. Naturally, we are aware that the definition of the elderly in terms of age might differ depending on a given region of the world and state organization. The majority of developed countries use the age of 65+ to define advanced-age individuals, which stems from their retirement programs with the benefit age of 60-65 years. Therefore, based on the above information and our analysis of the subject literature, we have adopted the age limit of 65 years.

Were all the scores performed by you based on the extracted data or at the time of hospitalization?

Thank you for the question. We would like to underline that our study was based on data extracted from the hospital’s database. The MEWS score was included in the database, as during the COVID-19 pandemic, the hospital used this very scale. That said, due to the fact that early warning scales are not commonly used in Poland, the scores on the other scales were calculated retrospectively based on parameters recorded on admission to the emergency department. We have added this information in the Data collection and measurements section.

Discussions

Give up adjectives of strengthening – own

Thank you for this comment. We have made a relevant correction following the Reviewer’s suggestion.

In the comparisons with other studies exclude the year of the study, only the author et al.

Thank you for this comment. We have made a relevant correction following the Reviewer’s suggestion.

References are placed at the end of the sentence.

Thank you for this comment. We have made a relevant correction following the Reviewer’s suggestion.

The discussions should also be completed with the other parameters presented in the results. The focus was only on age and scores (which is normal considering the purpose of the study), but the rest of the identified aspects should be discussed at least briefly.

Thank you for this suggestion. We have extended the Discussion accordingly.

Several studies indicate that advanced age is an independent factor associated with mortality in patients with COVID-19 - correct, but ethically when you refer to studies that associate age as a risk factor, your own study should not be taken into account.

Thank you for this remark. By referring to the previously published study, we wanted to underline the continuity of research that we conduct. However, we naturally understand the Reviewer’s reservations, thus we removed this publication from the literature list.

Some phrases should be rephrased, the duplication rate is quite high.

We have verified and changed.

Reviewer 2 Report

Comments and Suggestions for Authors

Thank you for the opportunity to review this paper. The authors describe the ability of different severity COVID-19 scores to predict in-hospital mortality. The authors use a large cohort, with in-depth analysis and proper methodology, and this subject is of major interest. Still, some issues should be addressed:

1.       The introduction include most of the relevant literature and explain the main reasons for the need of this study. I think that one of the major issues is who should be discharged from the ED and who to admit. A recent systematic review showed that discharging COVID-19 patients from the ED, even with severe disease, is safe with correct patient selection (PMID 38104299). I recommend the authors to address this aspect with the example mentioned above.

2.       Methods:

a.       How did you exclude pneumonia from other pathogens?

b.       How many patients were excluded in total?

3.       Results:

a.       Do you have data on AKI at ED presentation and COVID vaccinations? These factors were shown to be related to hospital mortality and could be important as additive variables to the scores presented.

b.       I think an important sub-analysis should be among subjects presenting to the ED without the need of oxygen or with only use of nasal cannula. This subgroup is of interest for clinical decision making, while obviously those with a need of higher method for oxygenation will be hospitalized and treated.

c.       In the paragraphs on the pairwise comparisons I don't think you should write all comparisons, just the 2-3 must significant or important, and the rest describe in words. These numbers already appear in the table 3.

4.       Discussion:

a.       You mention higher ICU rates (line 258), while I don’t think this data appear in the results section. Please revise.

b.       The authors only mention REMS as the best tool, while they should add their opinion on the reason for it and base it on prior literature that assessed the REMS score.

Comments on the Quality of English Language

Minor corrections are needed.

Author Response

Dear Reviewer,

We wish to thank you very much for the time you have taken to read and review our paper. We hope the changes we have made improved the overall quality of the paper in line with your expectations.

Thank you for the opportunity to review this paper. The authors describe the ability of different severity COVID-19 scores to predict in-hospital mortality. The authors use a large cohort, with in-depth analysis and proper methodology, and this subject is of major interest. Still, some issues should be addressed:

  1. The introduction include most of the relevant literature and explain the main reasons for the need of this study. I think that one of the major issues is who should be discharged from the ED and who to admit. A recent systematic review showed that discharging COVID-19 patients from the ED, even with severe disease, is safe with correct patient selection (PMID 38104299). I recommend the authors to address this aspect with the example mentioned above.

Thank you for the valid suggestion. The Introduction has been extended to include this information.

  1. Methods:
  2. How did you exclude pneumonia from other pathogens?

Due to the retrospective nature of the study, the exclusion of pneumonia caused by other pathogens was based on the diagnosis made and entered in the patients’ medical history. In the hospital that provided the data, differentiation was made on the basis of microbiological testing and chest CT scan.

  1. How many patients were excluded in total?

The total number of excluded patients was 2330.We have added information on the number of excluded patients with respect to individual criteria in the Study cohort and the eligibility criteria section.

  1. Results:
  2. Do you have data on AKI at ED presentation and COVID vaccinations? These factors were shown to be related to hospital mortality and could be important as additive variables to the scores presented.

Thank you for this important question. The database obtained did not include information on vaccinations. As for AKI, during data analysis we identified 11 patients with AKI, of whom 9 died. However, since this number is negligible, we did not include this aspect in the results.

  1. I think an important sub-analysis should be among subjects presenting to the ED without the need of oxygen or with only use of nasal cannula. This subgroup is of interest for clinical decision making, while obviously those with a need of higher method for oxygenation will be hospitalized and treated.

Thank you for this suggestion. In the study, we mainly focused on early warning scales. Due to the retrospective nature of the data, we are limited in terms of information on oxygen therapy, as we only have a variable describing whether a patient required oxygenation or not, without specification of the method by which this therapy was administered. Therefore, we used the term „Passive oxygen therapy” in the analysis, with respect to patients who received oxygen therapy either through a nasal cannula or an oxygen mask.

  1. In the paragraphs on the pairwise comparisons I don't think you should write all comparisons, just the 2-3 must significant or important, and the rest describe in words. These numbers already appear in the table 3.

Thank you for this comment. We have made a relevant correction following the Reviewer’s suggestion.

  1. Discussion:
  2. You mention higher ICU rates (line 258), while I don’t think this data appear in the results section. Please revise.

Thank you for this comment. Table 1 contains a comparison of ICU admission rates between survivors and non-survivors (2.42% vs.32.08%). We have revised the description in the Discussion.

  1. The authors only mention REMS as the best tool, while they should add their opinion on the reason for it and base it on prior literature that assessed the REMS score.

Thank you for this valid comment. We have addressed it in the Discussion.

Round 2

Reviewer 1 Report

Comments and Suggestions for Authors

Congratulation!

Author Response

Dear Reviewer,

We wish to thank you very much for the time you have taken to read and review our paper again.

Reviewer 2 Report

Comments and Suggestions for Authors

Thank you for the opportunity to review this paper once again. The authors have addressed all my previous comments and the article has significantly improved. Some minor issues exist.

First, if I understand correctly, all patients were admitted to a ward (not only visited the ED for a few hours). I find it surprising that more than 50% of hospitalized patients did not require any oxygen support. Did you have a protocol to hospitalize moderate severity patients just for observation? Another surprise is by the very high number of patients requiring HFNC (18%) upon ED presentation. Is it similar to the literature?

Second, I suggest adding a sub-analysis similar to what you did for above or under age 65 for patients requiring or not requiring any oxygen support. These two patient populations are highly distinct and such analysis could be of value.

Comments on the Quality of English Language

Non significant

Author Response

Thank you for the opportunity to review this paper once again. The authors have addressed all my previous comments and the article has significantly improved. Some minor issues exist.

Dear Reviewer,

We wish to thank you very much for the time you have taken to read and review our paper again.

First, if I understand correctly, all patients were admitted to a ward (not only visited the ED for a few hours). I find it surprising that more than 50% of hospitalized patients did not require any oxygen support. Did you have a protocol to hospitalize moderate severity patients just for observation? Another surprise is by the very high number of patients requiring HFNC (18%) upon ED presentation. Is it similar to the literature?

Thank you for your comment. We confirm that all patients included in the study were admitted to the Emergency Department. During the COVID-19 pandemic, the National Institute of Medicine of the Ministry of Interior and Administration was a so-called “Covid Hospital” to which all patients who had symptoms of infection or whose infection was confirmed by an antigen test performed by the Emergency Medical Services were sent. According to our knowledge, standard triage was carried out in the hospital in accordance with good practices in hospital emergency departments approved by the Ministry of Health in Poland. The final decision to admit the patient to the ward was always made by the doctor on duty.

Second, I suggest adding a sub-analysis similar to what you did for above or under age 65 for patients requiring or not requiring any oxygen support. These two patient populations are highly distinct and such analysis could be of value.

Thank you for your suggestion. We agree that the groups (requiring and not requiring oxygen support) differ significantly. In this study, we wanted to focus on the prognostic value of EWS and patient age. Additionally, we are in the process of developing a multicenter study that will analyze the need for oxygen therapy, and we do not want to publish data from the same hospital twice.